MINIREVIEW
# Extended Plant Metarhizobiome: Understanding Volatile Organic Compound Signaling in Plant-Microbe Metapopulation Networks

Waseem Raza,[a,b] Zhong Wei,[a] Alexandre Jousset,[a,b] Qirong Shen,[a] Ville-Petri Friman[a,c]

[a]Key Lab of Plant Immunity, Jiangsu Provincial Key Lab for Organic Solid Waste Utilization, National Engineering Research Center for Organic-based Fertilizers, Jiangsu Collaborative Innovation Center for Solid Organic Waste Resource Utilization, Nanjing Agricultural University, Nanjing, People's Republic of China
[b]Utrecht University, Institute for Environmental Biology, Ecology, and Biodiversity, Utrecht, The Netherlands
[c]University of York, Department of Biology, York, United Kingdom

**ABSTRACT** Plant rhizobiomes consist of microbes that are influenced by the physical, chemical, and biological properties of the plant root system. While plant-microbe interactions are generally thought to be local, accumulating evidence suggests that topologically disconnected bulk soil microbiomes could be linked with plants and their associated rhizospheric microbes through volatile organic compounds (VOCs). While several studies have focused on the effect of soil physicochemical properties for VOC movement, it is less clear how VOC signaling is affected by microbial communities themselves when VOCs travel across soils. To gain a better understanding of this, we propose that soil microbe-plant communities could be viewed as "metarhizobiomes," where VOC-mediated interactions extend the plant rhizobiome further out through interconnected microbial metapopulation networks. In this minireview, we mainly focus on soil microbial communities and first discuss how microbial interactions within a local population affect VOC signaling, leading to changes in the amount, type, and ecological roles of produced VOCs. We then consider how VOCs could connect spatially separated microbial populations into a larger metapopulation network and synthesize how (i) VOC effects cascade in soil matrix when moving away from the source of origin and (ii) how microbial metapopulation composition and diversity shape VOC-signaling between plants and microbes at the landscape level. Finally, we propose new avenues for experimentally testing VOC movement in plant-microbe metapopulation networks and suggest how VOCs could potentially be used for managing plant health in natural and agricultural soils.

**KEYWORDS** bulk soil microbiome, microbial metapopulation networks, long-distance communication, microbe-plant crosstalk, rhizosphere microbiome, volatile organic compounds

Plant-associated microbiomes have received considerable attention from scientists as key components of plant development, growth, and immunity (1). In particular, the rhizosphere microbiome (rhizobiome), defined as the microbes that are influenced by the physical, chemical, and biological properties of the plant root system, has been demonstrated to play important roles in plant growth, nutrition, pathogen suppression, and stress resistance (2–5). Traditionally, these belowground plant-microbiome interactions are considered local, occurring within the immediate vicinity of the plant roots (including root tissues) (5). However, increasing evidence suggests that belowground plant-microbiome interactions extend over longer distances in the soil matrix via volatile organic compounds (VOCs), which could potentially connect plant roots, rhizobiomes, and bulk soil microbiomes (6, 7). VOCs are a broad group of small lipophilic compounds ($<C15$) with low molecular weight (100 to 500 Da), high vapor

Address correspondence to Zhong Wei, weizhong@njau.edu.cn, or Ville-Petri Friman, ville.friman@york.ac.uk.

pressure, and low boiling point (7, 8). These characteristics allow VOCs to diffuse through gas- and water-filled pores, enabling a range of biological functions important for microbe-microbe and plant-microbe interactions (6, 8). For example, microbial VOCs can act as nutrient sources (9) and modulate plant vegetative growth, flowering, and immune responses (8). They can further trigger both antagonistic and synergistic interactions among plants, pathogens, and other soil organisms, such as nematodes and protists (10–13), and are important for competitive (antibiosis) and facilitative (cross-feeding) microbial interactions and microbiome assembly (14–16). Similarly, plant-produced root VOCs can act as antimicrobials, food sources, chemo-attractants or signaling chemicals (17, 18), affecting soil microbe community diversity, composition, and functioning (7). Soil VOC effects are thus omnidirectional and complex and have been shown to take place within and between different trophic levels (6).

While several studies have focused on the effect of soil physicochemical properties in governing VOC movement in the soil matrix (16–19), it is less clear how VOC signaling is affected by biotic interactions. Moreover, most of the previous and ongoing work has focused on cataloging the structure and ecological roles of VOCs under laboratory conditions, focusing mainly on interactions between pairs of organisms (7, 19). While this approach allows the controlled study of VOC mechanisms, findings are difficult to extrapolate on more natural and ecologically complex communities. For example, while it is well established that VOC effects can impact distant individuals or even populations (20, 21), it is unclear how VOCs travel through microbial metapopulations. In other words, only a little is known about how VOC signals change when blending with VOCs produced by adjacent populations, which could ultimately determine how VOC effects cascade in space when moving away from the source of origin in the soil. Similarly, while microbial community properties have been shown to drive VOC production locally (22, 23), it is not clear how microbial metapopulation composition and diversity shape VOC-signaling at the landscape level. Given the potential importance of VOCs for soil ecology and agricultural productivity (8, 24), it is important to start considering VOC signaling in plant-microbe communities over larger spatial scales (Fig. 1A). In this minireview, we provide an outlook on the nature and dynamics of VOC-mediated interactions, mainly focusing on soil microbial communities. We also propose a framework on how VOC effects could cascade through microbial metapopulation networks, potentially enabling an extended metarhizobiome by connecting plant roots, rhizobiome, and bulk soil into a cohesive underground ecosystem.

## VOCs ARE PRODUCTS OF LOCAL ENVIRONMENTS WITH POTENTIALLY GLOBAL EFFECTS

The microbial activity and plant roots are the main sources of VOCs in the soil (14, 19). Additionally, uptake of VOCs from the atmosphere, degradation of organic material, and application of organic fertilizers and irrigation contribute to the soil VOC pool (25, 26). Soils can also retain VOCs, and the patterns of adsorption and desorption depend on the type of VOCs and soil properties (27, 28). For example, VOC desorption from soils has been shown to increase with decreasing soil particulate size (29) and the number of carbon atoms present in the benzene ring of VOCs (30). Furthermore, VOC desorption tends to peak during periods of high temperatures and low moisture, suggesting that compounds accumulated during the winter may be released later in the summer, even after the sources of VOC emission have long vanished (31). Further, VOCs can escape to the atmosphere (32), bind to organic matter and mineral surfaces (33), undergo biotic and abiotic degradation (25, 26), dissolve in soil water solution (34) and get trapped in macro- and micropores in the soil (29; Fig. 1B). The movement of VOCs in soil results from diffusion and advection; diffusion is driven by concentration gradients, and advection can be driven by pressure, density, gravity, or thermal gradients (35, 36). The bulk water flow also plays a significant role in the movement of nutrients, organic waste, microbes, and VOCs in the soil (36, 37). Likewise, contiguous and interlocking channels formed in the soil through processes of desiccation, growth

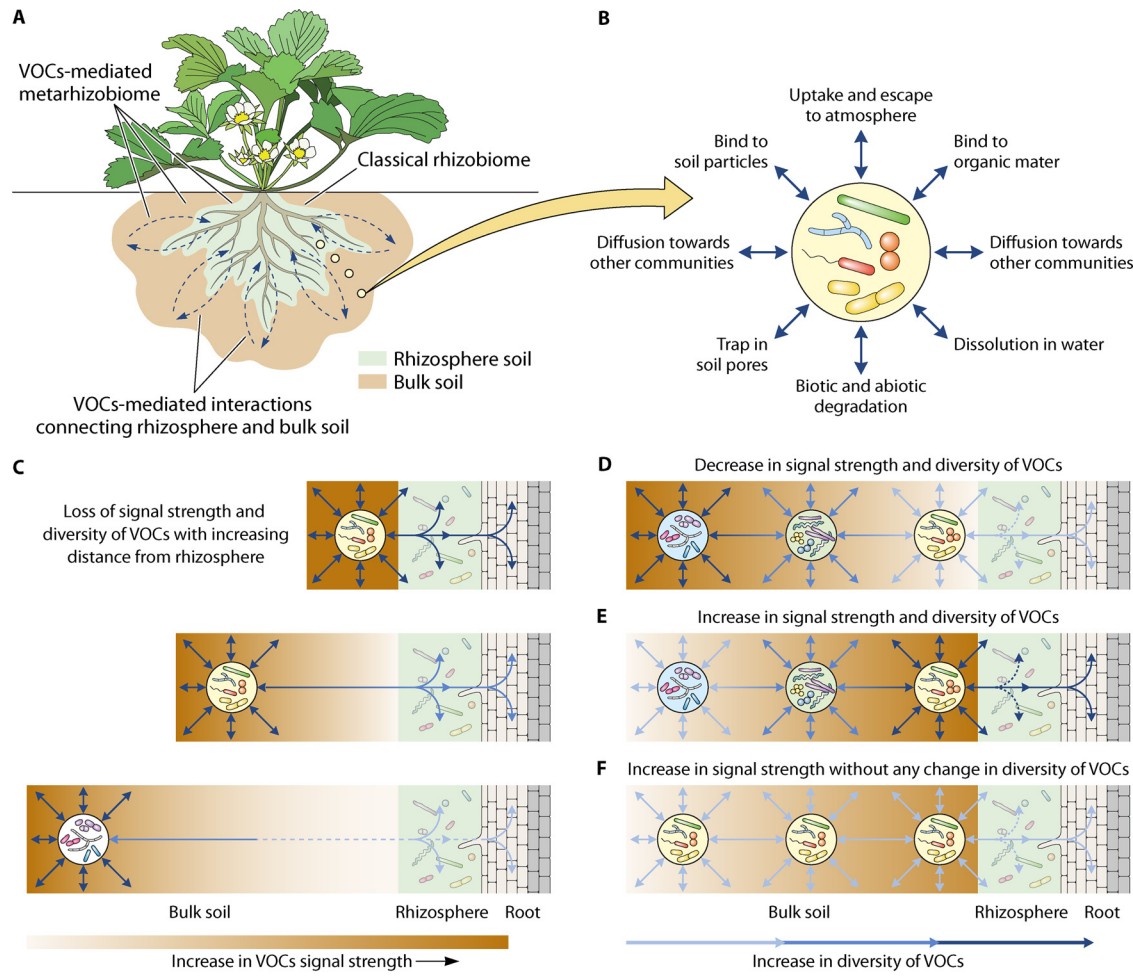

**FIG 1** Volatile organic compound (VOC)-mediated interactions can link plants with microbial metapopulation networks. (A) The classical rhizobiome is located in the close vicinity of the plant roots, while VOCs produced by microbes and plant roots disperse over long distances in the soil matrix, being able to connect and mediate multidirectional interactions among physically disconnected metapopulations of rhizosphere and bulk soil and plant (blue arrows). (B) A single microbial metapopulation in the bulk soil. The fate of emitted VOCs depends on the exchange rate and retention properties of VOCs, soil properties, and environmental conditions, which ultimately determine VOC movement, binding, evaporation, and dissolution. (C) VOC diffusion signal strength (amount of VOC) from and toward the bulk soil (VOC source) to rhizosphere soil as a function of physical distance; VOCs will have a stronger effect on the rhizosphere and plant roots when in close vicinity (top), and this effect will become weaker with increasing distance (middle and bottom). (D to F) Changes in the signal strength and diversity of VOCs between metapopulations in the bulk and rhizosphere soil. It is possible that both the signal strength and diversity of VOCs decrease as a function of distance from the source of origin (D). Alternatively, the original VOC signal could be strengthened when passing through similar metapopulations through "sequential community activation" (E), which could also further increase the diversity of VOCs as adjacent communities blend in their own VOCs (F). In panels C to F, the green color shows the VOC signal strength, and the shade of the blue arrows shows the diversity of emitted VOCs. The interaction described in panels C to F could also be initiated by plant root-emitted VOCs cascading toward nearby and distant communities in a similar fashion.

and decay of roots and mycelia, and burrowing animals act as superhighways for long-distance movement of VOCs (37, 38). Hence, soil physicochemical, environmental, and VOC properties are important in determining the adsorption-desorption dynamics and the effect radius of VOCs in the soil matrix.

While the rhizosphere gradient size for most biotic and abiotic processes has been reported as 0.5 to 4 mm and >20 mm for inorganic gases ($CO_2$ and $O_2$) (39, 40), there is no study explicitly testing the VOC diffusion dynamics in the soil. However, a few studies have explored the effect of distance on VOC-mediated interactions (7, 41). For example, a study conducted using an olfactometer system demonstrated that plant root VOCs can attract beneficial soil bacteria from as far as a 12-cm distance (7). In another field study, it was shown that nematodes can sense a root-produced terpene

VOC, (E)-$\beta$-caryophyllene, from a 50-cm distance (41). Moreover, diffusion experiments conducted at up to 12-cm distances using pure VOC standards suggest that their diffusion capability is specific to given VOC compounds (7, 24). Together, these findings suggest that VOCs can extend local microbiome interactions further out, potentially bidirectionally linking bulk and rhizosphere soils into a plant metarhizobiome. However, relatively much less is known about the significance of microbial interactions for the production and movement of VOCs in the soil.

## MICROBIAL INTERACTIONS WITHIN A POPULATION DETERMINE LOCAL VOC PRODUCTION

The production of VOCs is influenced by both abiotic and biotic microenvironmental conditions locally, which include intra- and interspecific microbial interactions (22, 23), substrate composition, temperature and moisture among others (42). Recent studies have demonstrated that competition between cooccurring species in a local population can increase the relative proportion of bioactive VOCs (22, 23, 43). For example, the production of antibacterial VOCs peaked at the intermediate community richness level in a synthetic 12-species model bacterial community (22). Interestingly, this effect coincided with high bacterial community evenness, which could have allowed more even VOC production by each individual species, and in support of this, the antibacterial activity of communities correlated positively with the number of produced antibacterial VOCs (22). Similarly, VOC effects are also affected by the absence of certain species as shown by another study where the loss of bacterial species was associated with reduced production and activity of antifungal VOCs (44). Such effects could be driven by taxa-specific VOC interactions, which have been shown to vary from positive to neutral and negative depending on the specific interacting species pair (22, 23). Moreover, it has been found that bacterial communities can produce "emergent" VOCs that cannot be detected when the VOC production is measured in bacterial monocultures (22, 23, 43). This could be because the bacterial metabolism is often changed in the presence of other species, which could trigger the upregulation of otherwise silent VOC metabolism-related genes (45). It has also been shown that pairwise VOC responses can be asymmetric. For example, VOCs produced by *Verticillium longisporum* fungi upregulated the metabolic activity of *Paenibacillus polymyxa*, while the VOCs of *P. polymyxa* inhibited the cellular metabolism and growth of *V. longisporum*, but upregulated genes related to stress responses and the production of antimicrobial VOCs (46). These findings suggest that VOCs could drive and be a result of potential coevolutionary dynamics that warrant further study in the future (24). Microbial VOC interactions are also likely to have indirect effects on other organisms, such as plants. Recently, it was demonstrated that bacterial communities that produce large amounts of bacterium-inhibiting VOCs produce a small amount of plant growth-promoting VOCs (22), which is indicative of a trade-off between functionally different classes of VOCs. Likewise, VOCs emitted by plant roots could indirectly affect microbial interactions within distantly located microbial populations. For example, insect-damaged maize roots change their VOC emission, leading to secretion of (E)-$\beta$-caryophyllene as the main VOC, which attracts entomopathogenic nematodes (12, 41). Similarly, tomato roots infected with the *Fusarium oxysporum* fungal pathogen have been shown to emit several VOCs with known antifungal activity (47), which suggests that plant pathogens could indirectly affect rhizosphere and bulk soil microbiomes by triggering changes in plant root VOC production. Together, the above-described evidence suggests that local VOC production is highly dependent on the microbial community composition and diversity and the specific interacting species.

## PREDICTING VOC SIGNALING IN MICROBIAL METAPOPULATION NETWORKS

While VOC-mediated interactions are well recognized, it is less clear how VOC effects cascade in space when moving away from the source of origin and how microbial metapopulation composition and diversity shape VOC-signaling at the landscape

level. The VOC effects are likely to decrease as a function of distance in the soil matrix. In support of this, Schulz-Bohm et al. (7) found a drastic decrease in the detectable amounts of VOCs with sampling distance from the source of origin in the soil. Hence, VOCs are likely to have relatively stronger effects on nearby communities (Fig. 1C and D), while communities located further away will be less affected due to the natural loss of VOCs over longer distances because of adsorption, trapping, degradation, and dissolution (25, 26, 28). Additionally, the original VOC signal could be amplified by adjacent microbial populations when moving away from the site of origin. It is known that different microbial species produce distinct sets of VOCs (43, 48), and their VOC production is affected by local microbial interactions and the surrounding environmental conditions (22, 23, 42). Moreover, airborne VOCs have been reported to alter soil microbial community composition (16), which is strongly correlated with the VOC emission profiles of "source" and "target" populations (49–51). Together, these findings support the concept that an initial VOC signal could blend with the VOCs emitted by adjacent microbial populations, leading to the amplification or complementation of the original signal and a potential increase in the total amount and diversity of emitted VOCs (Fig. 1E). While this could lead to a decrease in the relative concentrations of VOCs, the amplification of a specific VOC signal could also occur if the first VOC signal triggers the production of the same VOC by the adjacent community, potentially along with other VOCs, leading to 'sequential community activation' via amplification of the original VOC signal. In support of this, it has been shown that VOC profiles are more similar among closely related microbial species (48), and VOC emission has been found to correlate negatively with soil bacterial diversity (49). While more direct experimental evidence is needed, there is a possibility that taxonomically or functionally similar microbial populations could respond to conspecific signals in a similar way, leading to the amplification of the original VOC signal (Fig. 1F).

Current evidence also suggests that the specificity and bioactivity of VOCs is likely to further complicate VOC-signaling outcome in soils (19, 48). For example, schleiferon A VOC is formed via a nonenzymatic reaction, employing precursor VOCs (acetoin and 2-phenylethylamine) that could be emitted by microbes of the same or different species (52). In contrast to such generalist VOCs, microbes also produce specialist VOCs that are specific to certain microbial taxa (48). The taxon-specific VOCs could play a smaller role in spatially heterogenous soil communities if their signal is not received in the absence of specific "responder" species. In contrast, less-specific signals might get amplified more often, having potentially more far-reaching effects across microbial metapopulation networks. As a result, some VOCs could be functionally redundant (19, 28). Moreover, the VOC bioactivity and species VOC sensitivity will likely be important for VOC outcomes in the soil. For example, the same VOCs produced by bacteria can exert no or few effects on one fungal species (*Fusarium solani*) but at the same time showed a very high bioactivity to *Pythium* species (oomycetes) (53). VOC bioactivity could also be affected by the total amount of VOC produced. In support of this, soil VOC emission has been found to positively correlate with the abundances of prokaryotic *Bacteroidetes* and *Proteobacteria* phyla in one study (49) and with *Firmicutes*, *Proteobacteria*, *Actinobacteria*, and *Crenarchaeota* abundances in another study (50). These findings suggest that VOC production could be driven by density-dependent effects, where the most abundant taxa could have the strongest effect (54) on VOC-mediated signaling. Alternatively, VOC responses could be nonlinear, where only VOCs exceeding certain response thresholds, or highly bioactive VOCs (19, 22), would be able to influence adjacent microbial populations. In this case, taxa present in low relative abundances could be important contributors, as low concentrations of VOCs could mediate response cascades between adjacent microbial populations as has been demonstrated in the case of antifungal VOCs produced by rare soil bacterial taxa (44) and a relatively rare *Paenibacillus* sp. bacterium that strongly affected the production of VOCs by other much more abundant members of the bacterial community (15). While VOC signaling is further shaped by variation in abiotic microenvironmental conditions

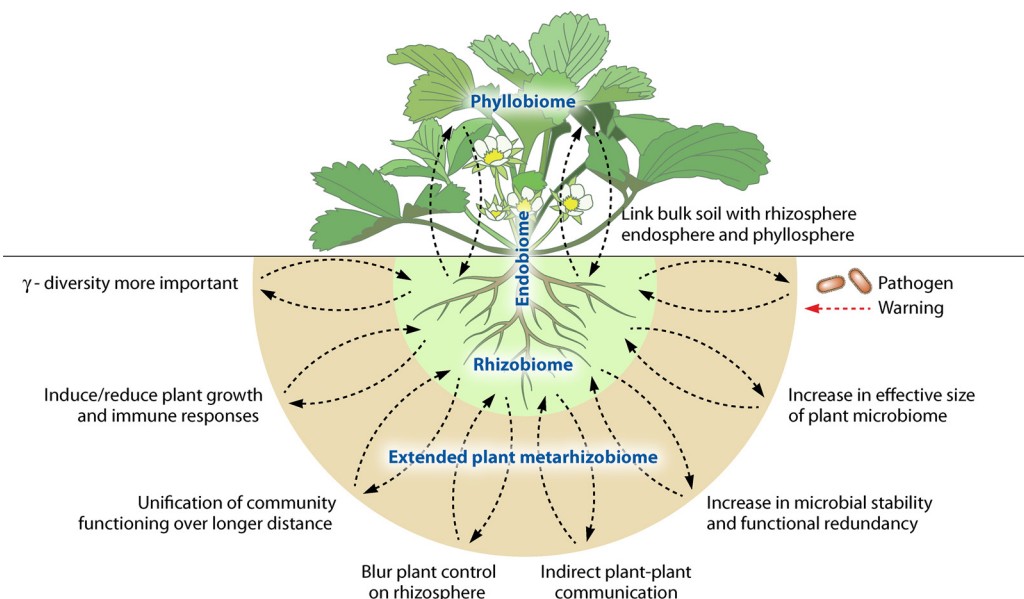

**FIG 2** Volatile organic compound (VOC)-mediated interactions between bulk soil and rhizosphere could be involved in a range of ecosystem-level functions and link the bulk soil microbiome with rhizobiome, endobiome, and phyllobiome. The metarhizobiome would allow the plant to connect physically larger space, diversity, and abundance of microbes in the soil matrix. Further, VOC effects emerging in the soil could cascade beyond the rhizosphere, affecting microbe-plant interactions inside the plant (endosphere) and on the plant leaves (phyllosphere), linking below- and aboveground microbiomes.

of a metapopulation (25), signal cascades might follow natural VOC diffusion in the soil matrix, creating subnetworks and feedback loops between certain "source" and "target" populations. In the future, the ideas presented above need to be experimentally tested to better understand the complex chemical interplay of VOCs in the soil matrix. This could be, for example, achieved by adopting aquatic metapopulation microcosm methods for soil systems (55).

## METARHIZOBIOME: LINKING MICROBIAL METAPOPULATION NETWORKS WITH PLANTS

Even though the bulk soil acts as an initial microbial pool for plant rhizobiome, the growth, development, and aging of plants cause clear shifts in rhizobiome composition, making it distinct from the bulk soil (56). As a result, bulk and rhizosphere soils have very dissimilar community structures, showing clear differences in the relative abundance of different bacterial taxa (57). These relative abundance differences are likely to be correlated with differences in bulk and rhizosphere soil VOC production profiles (49, 50). In further support of this, VOCs belonging to some chemical groups (i.e., alcohols, sulfur compounds, some ketones, and aromatic compounds) are predominantly produced by rhizosphere microbes compared to bulk soil microbes (58). Also, plant roots release VOCs (i.e., terpenoids, benzenoids, aliphatics, aromatics, fatty acids, etc.) into the rhizosphere (18, 47), making it chemically more diverse than the bulk soil. Rhizosphere soil is thus likely to be a hot spot for VOCs (5, 59), allowing plants to extend their rhizobiome into the bulk soil, while the effects from the bulk soil into the rhizosphere could be relatively weaker. The resulting metarhizobiomes would encompass a far larger space, resulting in a potentially larger number of interactions between a more diverse set of microbial taxa present in both rhizosphere and bulk soils (Fig. 2). Based on the current data on microbial abundance and distance-decay patterns in the soil microbial communities (60), increasing the interaction range from millimeters to centimeters (7) will considerably increase the effective size of the plant rhizobiome. This could potentially result in a large range of interactions across generally larger plant rhizobiome networks. Increasing the interaction network size could

mSystems®

also potentially have positive effects on plant rhizobiome stability if metarhizobiomes harbor greater species diversity and functional redundancy (61), being able to act as a source population if soil conditions change significantly, for example, during crop rotation or tillage (62).

The extension of the microbial interaction range into the bulk soil via VOCs could promote other long-distance signaling mechanisms, such as ion channel-mediated electrical signaling (63), potentially linking the activity of bulk soil communities with plant growth (Fig. 2). VOC-mediated signaling could also help plants to defend against pathogen attack. For example, in response to VOCs produced by the fungal pathogen *Fusarium culmorum*, the bacterium *Serratia plymuthica* has been shown to upregulate the production of sodorifen VOC (51), which induced the expression of plant defense-related genes in *Arabidopsis thaliana* (64). It is thus plausible that microbes are the first to sense the stress and produce specific metabolites to alert their host plant, as has also been suggested by Rizaludin et al. (65). VOC-sensing bacteria could thus warn plants about invading pathogens earlier by allowing activation of immune responses (VOC priming effect) in response to VOCs emitted by distant microbial communities (66), as has also been reported for aboveground VOC-mediated plant-to-plant warning against insect and disease attack (20). Furthermore, plants have also been reported to sense, integrate, and respond to plant-plant cues transmitted through roots (67, 68), which suggests that microbial populations could affect VOC signaling between adjacent plants (58). Similarly, plants could act as mediators and connect microbial metapopulations via VOCs, potentially leading to VOC-mediated interdependences and metarhizobiome stability at the landscape level, highlighting the importance of Gamma diversity. For example, Dharanishanthi et al. (69) reported that modification of the environmental pH by neighboring bacterial species could be used as a clue about nutrient availability by local bacteria, linking individual bacterial physiology to macroscale collective behavior.

Microbes residing in the soil can alter plant VOC profiles as has been reported for faba bean plants treated with arbuscular mycorrhizal fungi (70) and maize plants treated with the plant-beneficial bacterium *Pseudomonas putida* (71). Hence, considering long-distance VOC dispersion (7, 16, 38), it is plausible that VOC-mediated interactions triggered by bulk soil metapopulations could affect plants directly or create conflicts by blurring the boundary of plant control over the rhizobiome. Similarly, plant root VOCs can influence rhizosphere microbial community composition (16, 72), and this effect could extend to bulk soil microbiomes (7) either directly or indirectly via the rhizosphere microbiome. Plant-associated rhizobacteria can induce plant defenses against herbivores, while plants can, in turn, attract natural enemies of herbivores by emitting herbivore-induced plant VOCs (20). Similarly, plants can affect the rhizosphere microbiome of neighboring plants via rhizobacterium-induced aboveground plant VOC production (73). These findings support the concept that VOCs emerging in the bulk soil could also have effects beyond the rhizosphere, affecting the functioning of the whole-plant metamicrobiome, including endosphere and phyllosphere (Fig. 2). VOCs could thus potentially be important in linking plant below- and aboveground microbiomes.

## FUTURE PERSPECTIVES

To test our ideas and to develop a predictive theoretical framework on plant metarhizobiome functioning, much more experimental data are required. This could be achieved by developing highly trackable rhizobox and olfactometer systems that allow direct manipulation of VOC diffusion range, microbial community composition, and the abiotic environment in plant-microbe metapopulation networks. Moreover, a careful combination of complementary field approaches is needed to study the type and diffusion radius of VOCs of naturally distributed microbial populations in relation to soil physicochemical properties and climatic factors. The VOC-mediated interactions will not increase the total volume of the rhizosphere but will affect the metabolism

and physiology of (micro)organisms beyond the rhizosphere environment to at least the centimeter scale as suggested by de la Porte et al. (74). This could be especially important in the context of ongoing climatic change, allowing us to better understand how temperature and moisture drive the diffusion range of VOCs in the soil compared to soluble compounds. In addition to quantifying the range of VOC-mediated interactions, it will be important to compare the relative importance of different VOCs and their functional redundancy and diversity in microbial communities. For example, identifying potential keystone microbial species with relatively strong VOC-mediated interactions at the community level that could be used as microbial inoculants could be especially useful during intercropping periods. For example, choosing crop combinations based on VOC signal "compatibility" could be used as selection criteria for increasing agricultural ecosystem productivity. The manipulation of bulk soil microbiome could help to avoid conflicts with the plant and rhizobiome, potentially leading to higher functional stability and redundancy. Several VOCs can also be synthesized, making it potentially possible to apply them as transient and ecologically compatible biological control agents. Further, linking VOC patterns with metagenomic, transcriptomic, and metabolomics data could help to elucidate to what extent VOC production patterns can be predicted based on the genetic composition of microbial communities and if the underlying VOC pathways can be identified. A combination of existing and emerging omics and computational technologies could further help to identify chemical pathways underlying VOC production (24, 75). In addition, phenotyping of VOC emissions by using inexpensive small-scale trapping devices, smartphone-based VOC-sensitive sensors, and portable instruments for real-time measurements could help to better comprehend the dynamics of VOC emissions and discriminating genotype-specific and stress-related VOC profiles and patterns (76). Finally, while VOC-mediated interactions are known to have an important role in microbial ecology, they could also drive microbial evolution by selecting for VOC-resistant bacterial genotypes similar to soluble antimicrobial compounds (77) or facilitate other nutritional or stress-related adaptations. Proposed experimental model systems would allow testing such evolutionary questions and identifying genes and molecular mechanisms that play important roles in VOC interactions.

## CONCLUSIONS

Here, we propose that VOCs could coordinate bulk and rhizosphere soil microbiome functioning as a metarhizobiome, superseding the topological range limitation of contact-dependent microbe-microbe-plant interactions. Such plant metarhizobiomes would include microbes residing in the near physical vicinity of the plants (rhizobiome), as well as the VOC-connected populations located further apart in the bulk soil. Such multidirectional long-distance communication could fundamentally change how we perceive microbial ecology in the spatially structured soil matrix, allowing plant-microbe metapopulations to interact and trade information without restrictions imposed by the proximity and cooccurrence of the same local habitat. The attained knowledge could be potentially further used in the management of plant health in the agricultural context and to understand plant-microbe biodiversity and distribution in natural environments. The proposed predictions put forward by our conceptual framework should be rigorously tested in the future. This could be achieved by bringing together interdisciplinary scientists working on microbial ecology and evolution, genetics, biochemistry, and plant biology and by taking advantage of bespoke experimental systems that allow direct manipulation and quantification of microbe-plant communities and emitted VOCs.

## ACKNOWLEDGMENTS

W.R. receives funding from the European Union's Horizon 2020 research and innovation program under the Marie Skłodowska-Curie grant agreement no. 838710-ReproDev, the National Natural Science Foundation of China (no. 42007038), and Fundamental Research Funds for the Central Universities (no. KJQN202117). Z.W. is supported by the National Natural Science Foundation of China (no. 42090060,

41922053) and the National Key Research and Development Program of China (no. 2018YFD1000800). V.-P.F. is funded by the Royal Society (RSG\R1\180213 and CHL\R1\180031) and is jointly funded by a grant from UKRI, Defra, and the Scottish Government, under the Strategic Priorities Fund Plant Bacterial Diseases Program (BB/T010606/1).

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
