## [Reviewer comments · mSystems]

Extended Plant Metarhizobiome: Understanding Volatile Organic Compounds Signaling in Plant- Microbe Metapopulation Networks

Waseem Raza, Wei Zhong, Jousset Alexandre, Shen Qirong, and Friman Ville-Petri

Corresponding Author(s): Wei Zhong, College of Resources and Environmental Sciences, Nanjing Agricultural University

Review Timeline:

Submission Date:	June 29, 2021
Editorial Decision:	July 8, 2021
Revision Received:	August 4, 2021
Accepted:	August 9, 2021

Editor: Heather Bean

Reviewer(s): Disclosure of reviewer identity is with reference to reviewer comments included in decision letter(s). The following individuals involved in review of your submission have agreed to reveal their identity: Scott D. Soby (Reviewer #1)

Transaction Report:

DOI: <https://doi.org/10.1128/mSystems.00849-21>

Prof. Wei Zhong
College of Resources and Environmental Sciences, Nanjing Agricultural University
Nanjing
China

Re: mSystems00849-21 (Extended Plant Metarhizobiome: Understanding Volatile Organic Compounds Signaling in Plant-Microbe Metapopulation Networks)

Dear Prof. Wei Zhong:

Reviewer comments are found at the end of this letter.

Your minireview is likely to be accepted once the indicated changes are made.

Author Bios: If you would like a brief biographical sketch of each author (limit, 150 words) to be published at the end of your article, please submit text and photos with your modified manuscript. Please refer to the instructions posted at <https://journals.asm.org/journal/msystems/article-types>

Figures **1 and 2** in your manuscript are well designed, but may still benefit from graphical enhancement. We now offer our authors the services of ASM's contracted artist, Patrick Lane of ScEYence Studios. This art enhancement service is free of charge to authors of minireviews and full-length reviews, and turnaround time is fast. If you are interested in this service, please contact Patrick on receiving this letter. Complete contact information for Patrick and further instructions are posted at <https://journals.asm.org/pb-assets/pdf-text-excel-files/graphical-enhancement-support.pdf>

Please return your modified manuscript within 60 days; if you cannot complete the modification within this time period, please contact me. If you decide that you do not want to modify the manuscript and wish to submit it to another journal, please notify me of your decision immediately so that the manuscript can be formally withdrawn.

To submit the modified manuscript, log onto the eJP submission site at <https://msystems.msubmit.net/cgi-bin/main.plex>. If you cannot remember your password, click the "Can't remember your password?" link and follow the instructions on the screen. Go to Author Tasks and click the appropriate manuscript title to begin the resubmission process. The information you entered when you first submitted the paper will be displayed. Please update the information as necessary. Provide (1) point-by-point responses to the issues raised by the reviewers as file type "Response to Reviewers," not in your cover letter, and (2) a PDF file that indicates the changes from the original submission (by highlighting or underlining the changes) as file type "Marked Up Manuscript - For Review Only."

To submit your modified manuscript, log onto the eJP submission site at <https://msystems.msubmit.net/cgi-bin/main.plex>. If you cannot remember your password, click the "Can't remember your password?" link and follow the instructions on the screen. Go to Author Tasks and click the appropriate manuscript title to begin the resubmission process (ONLY the

corresponding author will have access to the full record for resubmission). The information that you entered when you first submitted the paper will be displayed. Please update the information as necessary and do the following:

- 1) Provide point-by-point responses to the issues raised by the reviewers in a file designated as "Response to Reviewers" (NOT the cover letter).
- 2) Upload ALL of your source files (not PDF and not just the files requiring modification) and make sure that all elements meet the technical requirements for production.
- 3) Do not provide a highlighted or tracked-changes copy of the paper in the main manuscript upload. This should be a clean copy instead. You may provide the compare copy separately by uploading it as a "Marked-Up Manuscript" file.
- 4) Make sure that the figure legends are included in the main manuscript file (not uploaded separately).

If you would like to submit an image for consideration as the Featured Image for an issue, please contact mSystems staff.

Sincerely,

Heather Bean
Editor, mSystems

Journals Department
Reviewer comments:

Reviewer #1 (Comments for the Author):

The paper is much improved with this draft. The expansion of references to support the presented ideas was just what was needed.

There are a number of minor grammatical errors that should be cleaned up, such as agreement of number between subject and verb, and choice of prepositions to clarify the relationship between clauses. Details are provided in the attachment.

Reviewer #2 (Comments for the Author):

The authors have addressed most of the comments raised by the referees. The authors have softened several of the hypotheses proposed in the previous version of the manuscript and have included a number of new references that provide experimental support to the hypotheses

presented in this mini-review. The manuscript reads well and I only have minor comments on this manuscript.

- Line 52: pathogenic microorganisms have not been listed within the rhizosphere microbial community.

- Line 92: Change "VOCS" to "VOCs"

- Line 298-299: As indicated in previous evaluation reports, VOCs emission by plants and/or rhizospheric (micro)organisms will not cause an increase in the total volume of the rhizosphere, but affect the metabolism/physiology of (micro)organisms beyond the rhizosphere environment.

Third review of Raza et al.

L52. Do references 2-4 cover suppressive soils? I don't have these references handy, but don't see anything that obviously supports microbial disease suppression. There are hundreds of references to choose from.

L196. Nothing can be 'more unique' or less unique. If it is unique, it is the only thing like it.

L212-213. 'In this case, also **the rare microbiome (taxa present in low relative abundances)** could be important as **already** small concentrations...' Should be simplified to 'In this case, **taxa present in low relative abundances** could be important **contributors** as **low** concentrations...'

L243-244. I suggest changing, 'This could potentially allow **myriad of** interactions and generally larger plant-rhizobiome networks.' to 'This could potentially result **in a large range of interactions across** generally larger plant-rhizobiome networks.'

L273-275. Suggestion: 'blurring the plant control on the rhizobiome' to 'blurring the **boundary of** plant control over the rhizobiome'. Similarly, plant root VOCs can **influence** rhizosphere microbial community composition (15, 72) and this effect could extend to...'

L277-278. '**while plants can in turn attract natural enemies of herbivores by emitting herbivore-induced plant VOCs**'

L299-301. I suggest changing 'For example, identifying potential keystone microbial species that have relatively strong VOC-mediated interactions at the community level could be **potentially used as microbial inoculants, while VOC-mediated interactions could be especially important in intercropping.**' to 'For example, identifying potential keystone microbial species **with** relatively strong VOC-mediated interactions at the community level **that** could be **used as microbial inoculants could be especially useful during intercropping periods.**'

L306. It is highly unlikely that any naturally-produced VOC could be used as a biofumigant, in part because of some of the points you have made in the manuscript, but mostly because anything toxic enough to fumigate a soil environment is likely to be nondiscriminating in its killing, and if not so toxic, then unlikely to eliminate pathogens. In our experience VOCs are static agents rather than cidal, so once the agent is removed one would expect rapid recovery of the target populations. However, the idea has merit not as a fumigant but perhaps as a way to give seedlings a chance to establish in the presence of seedling disease fungi. Perhaps changing 'biofumigant' to 'a transient and ecologically compatible biological control agent' or 'an ecologically compatible stimulant of the plant's immune system' would suffice.

L310-311. Why not just say something like, '...a combination of existing and emerging omics and computational technologies' rather than giving a list that is likely to be soon outdated or that leaves out an important 'omics', like 'volatilomics'?

L337. The sentence doesn't make sense. I think the wrong word was deleted.

August 9, 2021

Prof. Wei Zhong
College of Resources and Environmental Sciences, Nanjing Agricultural University
Nanjing
China

Re: mSystems00849-21R1 (Extended Plant Metarhizobiome: Understanding Volatile Organic Compounds Signaling in Plant-Microbe Metapopulation Networks)

Dear Prof. Wei Zhong:

Your manuscript has been accepted, and I am forwarding it to the ASM Journals Department for publication. For your reference, ASM Journals' address is given below. Before it can be scheduled for publication, your manuscript will be checked by the mSystems senior production editor, Ellie Ghatineh, to make sure that all elements meet the technical requirements for publication. She will contact you if anything needs to be revised before copyediting and production can begin. Otherwise, you will be notified when your proofs are ready to be viewed.

As an open-access publication, mSystems receives no financial support from paid subscriptions and depends on authors' prompt payment of publication fees as soon as their articles are accepted. =

Publication Fees:

We recognize that the video files can become quite large, and so to avoid quality loss ASM suggests sending the video file via <https://www.wetransfer.com/>. When you have a final version of the video and the still ready to share, please send it to Ellie Ghatineh at eghatineh@asmusa.org.

Sincerely,

Heather Bean
Editor, mSystems

Journals Department
Phone: 1-202-942-9338